# The Acute Effects of Oral Administration of Phytic Acid-Chitosan-Magnetic Iron Oxide Nanoparticles in Mice

**DOI:** 10.3390/ijms20174114

**Published:** 2019-08-23

**Authors:** Norain Mohd Tamsir, Norhaizan Mohd Esa, Nurul Husna Shafie, Mohd Zobir Hussein, Hazilawati Hamzah, Maizaton Atmadini Abdullah

**Affiliations:** 1Department of Nutrition and Dietetics, Faculty of Medicine and Health Sciences, Universiti Putra Malaysia, 43400 Serdang, Selangor, Malaysia; 2Research Centre of Excellent, Nutrition and Non-Communicable Diseases (NNCD), Faculty of Medicine and Health Sciences, Universiti Putra Malaysia, 43400 Serdang, Selangor, Malaysia; 3Laboratory of Molecular Biomedicine, Institute of Bioscience, Universiti Putra Malaysia, 43400 Serdang, Selangor, Malaysia; 4Laboratory of UPM-MAKNA Cancer Research, Institute of Bioscience, Universiti Putra Malaysia, 43400 Serdang, Selangor, Malaysia; 5Materials Synthesis and Characterization Laboratory, Institute of Advance Technology (ITMA), Universiti Putra Malaysia, 43400 Serdang, Selangor, Malaysia; 6Department of Veterinary Pathology and Microbiology, Faculty of Veterinary Medicine, Universiti Putra Malaysia, 43400 Serdang, Selangor, Malaysia; 7Department of Pathology, Faculty of Medicine and Health Sciences, Universiti Putra Malaysia, 43400 Serdang, Selangor, Malaysia

**Keywords:** inositol hexaphosphate, magnetic nanoparticles, nanotoxicology, polymer-based nanocomposite

## Abstract

A nanocomposite, phytic acid-chitosan-magnetic iron oxide nanoparticles (IP_6_-CS-MNPs) has been used to treat colon cancer in vitro, previously. However, its potential toxicity in vivo has yet to be elucidated. Hence, the present study aimed to evaluate the acute effects of oral administration of IP_6_-CS-MNPs in mice. In this study, 1000 and 2000 mg/kg body weight (b.w) of IP_6_-CS-MNPs were orally administered to two different groups of BALB/c mice, once. Additionally, the mice in the control group were given only deionized water. After 14 days of post-IP_6_-CS-MNPs administration, in a similar way to the untreated mice, the treated mice showed no sign of mortality and abnormalities. However, the serum urea level of mice receiving 2000 mg/kg b.w of IP_6_-CS-MNPs was significantly higher than the control group (*p* < 0.05). The mice that received 1000 mg/kg IP_6_-CS-MNPs showed a significantly higher level of serum alkaline phosphatase (ALP) compared to the control group. However, there were no significant histopathological changes seen in the liver and kidneys of treated mice compared to the untreated group.

## 1. Introduction

Nanotechnology application in biomedicine has revolutionized dynamically. It has triggered a plethora of new magnetic iron oxide nanoparticles’ (MNPs) formulations and productions. Based on contrast enhancement agent in imaging, MNPs are now widely used in disease treatment due to its unique properties including biocompatibility, stability, being environmentally safe, as well as cheaply produced [1,2]. Nonetheless, MNPs have low solubility and tend to agglomerate due to the attraction to each other by groups of hydroxyl. This will subsequently increase the size of the nanoparticles and obstruct blood vessels [3]. Chemical co-precipitation is commonly used to synthesize MNPs because it is convenient and cost-effective [4]. This method of synthesis results in spherical shape and small size of nanoparticles [5]. There are several factors that may interfere with the magnetic properties, particles size, as well as the shape of MNPs via the co-precipitation method. Such factors may include the Fe^2+^ (ferrous)/Fe^3+^ (ferric) ratio, reaction temperature, the order of reactants’ addition, and the use of dispersing agent, [6] which usually results in nanoparticle agglomeration [7]. It has been revealed that the modification of MNPs’ surface by polymers such as chitosan enhances the instability between particles and prevents agglomeration [8].

Natural and synthetic biodegradable polymers as nanocarriers have been expansively applied in nanomedicine. They function as controlled release vehicles for various drugs, agents, peptides, proteins, etc. [9]. Polymer-based nanoparticles are less toxic and deliver efficiently [10]. They are also able to distinguish between cancer and healthy tissues and may help curb multidrug resistance [11]. That being said, chitosan has been considered as one of the best natural polymer-based carriers for the MNPs drug delivery system. Chitosan is a biodegradable and biocompatible polymer, derived from the chitin of fungi and crustacean shells. It exhibits distinctive biological activities such as antioxidant, anti-fungal, antibacterial, and anti-tumor [12]. The positive charge of chitosan also drives it to the negatively-charged cell membrane. Additionally, its mucoadhesive behavior prolongs the duration of blood circulation [13].

Conventional chemotherapy is considered effective. However, selective chemotherapeutic agents are known to have caused adverse effects, multiple drug resistance, and sometimes off-target actions [14]. Therefore, several approaches have been introduced to minimize the side effects and maneuver the actions of the drugs or agents towards the periphery of the targeted site. Phytic acid is a naturally occurring chemotherapeutic agent, which has been extensively studied for pharmacological properties. Phytic acid possesses a C_6_H_18_O_24_P_6_ (Figure 1) chemical formula known as inositol hexaphosphate (IP_6_). Inositol hexaphosphate is a saturated cyclic, which is ubiquitously found in the plant tissues and acts as storage forms of phosphorus in mammalian cells. IP_6_ in bran is abundantly found as a by-product of the rice-milling industry in Malaysia where rice is the staple food. IP_6_ has been found to have significant antioxidant [15] and anti-cancerous properties in different cell lines [16,17,18] as well as in in vivo models [19].

In 2017, Barahuie et al. [21] synthesized a phytic acid-chitosan-magnetic iron oxide nanocomposite (IP_6_-CS-MNPs) and found that it was not toxic to the normal cells (3T3 fibroblast cells). A recent work by Tan and colleagues [22] showed that IP_6_-CS-MNPs induced cell cycle arrest at G_0_/G_1_ phase and apoptosis in colorectal cancer cell lines. This proved that the nanocomposite is a potential anticancer agent. However, animal studies should be conducted before clinical studies to examine the potentially toxic effect of the newly-developed nanocomposite in a more complex system, which mimics human reactions. Therefore, the present study was designed to elucidate the acute oral toxicity of the IP_6_-CS-MNPs nanocomposite in mice.

## 2. Results

### 2.1. Animal Observation, Mortality, and Body Weight Change

An acute toxicity study using 18 female Balb/c mice (6–8 weeks) was conducted with an oral administration of 1000 and 2000 mg/kg body weight (b.w) of IP_6_-CS-MNPs (in 0.5 mL deionized water) to two different treatment groups, respectively. Another group (control group) received only deionized water (0.5 mL). The first group (controlled group) was only given 0.5 mL of deionized water. For 14 days the mice did not exhibit behavioral toxicity symptoms such as bleeding, vomiting, abnormal posture, diarrhea, irritation, breathing difficulties, restlessness, and mortality, despite being treated with 1000 or 2000 mg/kg of b.w of IP_6_-CS-MNPs. Therefore, the results proved that the median lethal dose (LD_50_) of IP_6_-CS-MNPs was greater than 2000 mg/kg of b.w. The body weight of the mice in the controlled and treatment group pre-experiment, and on the 7th and 14th day of post-experiment was recorded (Table 1). The body weight of the mice in the first week of the post-nanocomposite administration resulted in a slight decline. Nevertheless, after 2 weeks, the mice demonstrated slight weight gain.

### 2.2. Relative Organ Weight

The weight of mice organs (liver, kidney, spleen, brain, heart, colon, and lungs) from all groups of mice were used to calculate the relative organ’s weight. The results are recorded in Table 2. There was no significant difference in the relative organ’s weight between controlled and treatment groups (*p* > 0.05) on the 14th day of exposure to 1000 or 2000 mg/kg b.w of IP_6_-CS-MNPs.

### 2.3. Biochemical Analysis

Table 3 shows the biochemical test results obtained from mice serum after 14 days of IP_6_-CS-MNPs treatment. Liver enzymes such as alkaline phosphatase (ALP), aspartate aminotransferase (AST), and alanine transferase (ALT) were measured to assess the liver function whereas creatinine and urea were measured to evaluate kidney function. There was an imperative higher level of ALP in 1000 mg/kg b.w IP_6_-CS-MNPs-treated group compared to the other groups (*p* < 0.05). The serum urea level of mice treated with 2000 mg/kg of IP_6_-CS-MNPs was significantly higher (*p* < 0.05) than the controlled group and 1000 mg/kg of IP_6_-CS-MNPs. Nevertheless, the level of serum AST, ALT, and creatinine showed no significant difference between groups (*p* > 0.05).

### 2.4. Histopathological Evaluation

Toxicological lesions in the liver such as inflammation, necrosis, and regeneration were examined and scored. Figure 2 proved that the histopathological examination on the livers of treated mice revealed no remarkable lesions that could be attributed to the effect of 1000 or 2000 mg/kg b.w of IP_6_-CS-MNPs (Figure 2).

Besides that, the toxicological lesions in the kidney tissues such as inflammation, cellular cast, granular cast, protein cast, hydropic degeneration, and necrosis were examined and scored. Additionally, there were no histopathological changes observed in the kidney tissues (Figure 3). Therefore, both histopathological scorings of kidney and liver showed no remarkable lesions that could be attributed to the effect of 1000 and 2000 mg/kg b.w of IP_6_-CS-MNPs administration.

## 3. Discussion

Nanoparticles vary in shape and size. Hence, our previous study focused on the characterization of MNPs to prove that they were spherical with an average size of 15 *±* 6 nm [21]. Due to their tiny size, studies on their toxicity effects are crucial. Therefore, acute toxicity studies have been conducted on experimental animals and humans to evaluate the effects of single high dose and divided doses of test substances via multiple routes of administration in a short period of time [23]. This was evaluated based on several signs and symptoms such as abnormal posture or behavior and episodes of seizures. According to the Organization for Economic Co-operation and Development (OECD), it is imperative to observe the mortality rate, specifically the median lethal dose (LD_50_) as it is significant for the acute toxicity evaluation [24]. Previous work has claimed that the LD_50_ of uncoated iron oxide nanoparticles was 300–600 mg Fe/kg b.w [25]. Meanwhile, the current study discovered that there were no signs of toxicity and mortality throughout the experiment. Therefore, it can be suggested that the IP_6_-CS-MNPs LD_50_ is greater than 2000 mg/kg b.w.

Changes in experimental animals’ body weights could also be an important marker to evaluate their health status. Therefore, a loss of 10% or more of body weight in comparison to the initial body weight signals unwanted effects of the chemical substances [26]. The present study revealed a pattern in the weight of the mice from both groups. In the beginning, there was an insignificant decrease however, and it soon increased until the end of the experiment (*p* > 0.05). These changes in weight are concordant to a study conducted by Monika et al. [27]. Hence, the loss in body weight is related to the body’s normal physiological response as it gets adjusted to the exposure of chemical substances. This might be due to the decrease of food or water intake (appetite suppression), which alters the taste or odor of the orally administered IP_6_-CS-MNPs [28] or damage of major organs such as the stomach, intestine, kidneys, and liver [29].

Toxicity can also be detected based on the weight of the internal organs. In the present experiment, the internal organs’ weight including liver, kidneys, brain, lungs, spleen, and heart were measured and compared to the animal’s final body weight. Metabolic reactions caused by test drugs or chemicals may affect major internal organ weights. For example, an increase in the weight of the liver may indicate signs of hepatic hypertrophy, which is stimulated by hepatic enzymes [30]. Nevertheless, findings proved no significant changes in the relative organ weight. Therefore, it is possible that the nanocomposite may not elicit any deleterious effects on the organs of the mice.

In the study of toxicology, various blood analyses have been used to measure organ and cell damage, as well as the activation or inhibition of certain enzymes. Moreover, it is crucial to evaluate the liver and kidneys when it comes to the process of assessing the levels of toxicity in drugs or chemicals because they play a vital role in metabolic detoxification and excretion. For the liver, the level of certain enzymes such as alanine aminotransferase (ALT), alkaline phosphatase (ALP), and aspartate aminotransferase (AST) contained in the serum should be measured. According to Mahmudul et al. the reference range of ALP is between 30 and 130 IU/L, AST is between 50 and 150 IU/L, and ALT is between 10 and 40 IU/L [31]. The results showed a substantial increase in the ALP serum in treating the mice with 1000 mg/kg b.w of IP_6_-CS-MNPs (*p* < 0.05). Elevated reading of the ALP enzyme might be attributed to many reasons. Besides that, the ALP enzyme is also found in other tissues such as the bone, kidney, intestine, and placenta. Furthermore, the ALP level can help detect hepatobiliary diseases, osteoporosis, and fatty liver disease. [32]. The accumulation of ALP in serum from liver, bone, kidney, and intestine might result in the elevation of this enzyme. That being said, the long-term effects following the consumption of IP_6_-CS-MNPs needs further investigation.

Additionally, the levels of creatinine and urea in the serum can detect certain abnormalities in the kidneys. Results prove that the serum creatinine in the control and treated groups of mice did not show any significant difference. However, there was an increase in the level of serum urea in the group of mice treated with 2000 mg/kg b.w of IP_6_-CS-MNPs (*p* < 0.05) compared to the control group. Urea is the major nitrogenous waste product of metabolism and is generated from protein degradation in the body. According to the previous study, if the level of creatinine is normal but the level of urea in the blood has increased, then it might be due to dehydration, which results in reduced blood flow to kidneys, gastrointestinal bleeding, or high protein intake [33].

The histopathological assessments of both organs provide supportive evidence for biochemical analyses. The present study work revealed no significant abnormalities or lesions were seen in both liver and kidneys of IP_6_-CS-MNPs treated and control groups. In the future, it would be of great significance if studies may look at the IP_6_-CS-MNPs cell tissue and their potential neurotoxicity effects.

## 4. Materials and Methods

### 4.1. Chemicals and Reagents

Iron (III) chloride hexahydrate (FeCl_3_·4H_2_O, >99.8% purity) and iron (II) chloride tetrahydrate (FeCl_2_·6H_2_O) were acquired from Merck KGaA (Darmstadt, Germany). Phytic acid sodium salt hydrate with a molecular weight of 660.04 kDa and chitosan were purchased from Sigma-Aldrich (St. Louis, MO, USA). Ammonia was supplied from Scharlau (Sentmenat, Barcelona, Spain). Acetic acid was obtained from Hamburg Industries (Hamburg, Germany).

### 4.2. Magnetic Iron Oxide Nanoparticle Synthesis

The MNPs were prepared via chemical precipitation based on the method suggested by Barahuie et al. [21]. Initially, the mixture was made up of 2.43 g ferric chloride hexahydrate (FeCl_3_·4H_2_O), 0.99 g of ferrous chloride tetrahydrate (FeCl_2_·6H_2_O), and 80 mL of deionized water. Then it was topped with 6 mL drops of 25% ammonia. The mixture was maintained at a pH value of 10. Then, the mixture was sonicated for an hour at room temperature. Next, the mixture was centrifuged, and the obtained magnetic nanoparticles were repeatedly washed with deionized water to eliminate the traces of ammonia. The obtained magnetic nanoparticles (Fe_3_O_4_) were dried in the oven at 60 °C and grounded into powder for further characterization and analysis.

### 4.3. Coating of Magnetic Iron Oxide Nanoparticles with Chitosan

Approximately, 1 g of chitosan was dissolved in deionized water with 0.5% acetic acid and stirred for 3 h. The chitosan solution was then added to the MNPs suspension and stirred with a rotational speed of 600 rpm for 18 h at room temperature. The mixture was centrifuged and repeatedly washed with deionized water to remove traces of ammonia [34]. The product was then labeled as CS-MNPs.

### 4.4. Phytic Acid-Chitosan-Iron Oxide Nanocomposite Synthesis

Next, 2% of the phytic acid solution was added to the CS-MNPs. The mixture was vigorously stirred for 24 h. Finally, the mixture (labeled as IP_6_-CS-MNPs) was centrifuged, washed, and dried in the oven at 60 °C [21].

### 4.5. Experimental Animals and Grouping

A total of 18 female, Balb/c mice (6–7 weeks old) with an average weight of 18–20 g were obtained from Animal Resource, Faculty of Veterinary Medicine, Universiti Putra Malaysia (UPM), Serdang, Selangor. The procedures of mice treating were approved by the Institutional of Animal Care and Use Committee (IACUC), UPM. Its reference number was UPM/IACUC/AUP-R030/2016. Prior to the treatment, all animals were acclimatized for a week and were provided free access to standard mouse pellets and water. Animals were maintained under standard laboratory conditions. For example, the were kept under a 12-h light/dark cycle at a temperature of 25 ± 2 °C and in relative humidity between 30% and 70%. The mice were then randomly divided into three groups post-acclimatization (Table 4). On day one, the mice were given either deionized water or IP_6_-CS-MNPs via oral gavage. The IP_6_-CS-MNPs suspensions were sonicated before they were given to the mice in the treatment group.

### 4.6. Animal Observation

The animals were observed for any clinical signs of toxicity or possible mortality within the first 30 min, periodically within the first 24 h post-dosing and thereafter, daily for a total of 14 days. The body weight of the mice was measured before nanocomposite administration and weekly thereafter. 

### 4.7. Serum Collection

On the 15th day the mice were euthanized via exsanguination under ketamine (80 mg/kg) and xylazine (10 mg/kg) mixture. Prior to the euthanasia, the mice were allowed to fast for 3–4 h. Blood samples were then collected via cardiac puncture. The study used a 26-gauge needle and a 1 mL syringe under anesthesia which was stored in ice before serum separation. Serum samples were collected after centrifugation at a speed of 3000 rpm at 4 °C for 15 min. The samples were then kept at −20 °C until serum biochemistry analysis was conducted. 

### 4.8. Serum Biochemistry Analysis

The collected serum samples measured the levels of ALT, AST, and ALP to detect hepatotoxicity, urea, and creatinine for the evaluation of nephrotoxicity. The tests were done via the automated biochemistry analyzer (Tokyo Boeki Machinery Ltd., Japan).

### 4.9. Organ Collection 

Post mortem examination was done instantaneously after the sacrifice to identify abnormalities on the animals’ vital organs. Vital organs such as the liver, kidney, heart, lung, brain, spleen, and colon were harvested and washed with normal saline. They were then weighed to calculate the relative organ weight. The algorithm is as follows: relative organ weight = (organ weight/ body weight) × 100.

### 4.10. Histopathological Assessment

The livers and kidneys were fixed in neutral-buffered formalin and were embedded in paraffin wax for the histopathological analysis. After cooling, the paraffin blocks were divided into sections with 5 µm of thickness, which was mounted onto glass slides, deparaffinized, and stained by hematoxylin and eosin (H&E). Slides were viewed under a light microscope to observe abnormalities. Besides that, the light microscope was able to score the liver and kidney tissues to determine the changes that occurred in the experimentally-induced histopathologic parameters. 

### 4.11. Statistical Analysis

The experiments were expressed as the mean and standard error of mean (SEM) values for each group. The data were analyzed using the one-way analysis of variance (ANOVA) with Tukey’s test to evaluate the differences between groups. Statistical analyses were performed by IBM SPSS Statistics, Software version 25 (SPSS Inc., Chicago, IL, USA). As a result, *p* < 0.05 was considered to be significantly different.

## 5. Conclusions

We conclude that the LD_50_ of the IP_6_-CS-MNPs is greater than 2000 mg/kg of b.w and caused a slight biochemical alteration after oral administration. Nevertheless, there were no mortality and histopathological changes seen. Further investigations of the chronic toxicity and biodistribution should be conducted for a detailed safety profile on IP_6_-CS-MNPs.

## Figures and Tables

**Figure 1 ijms-20-04114-f001:**
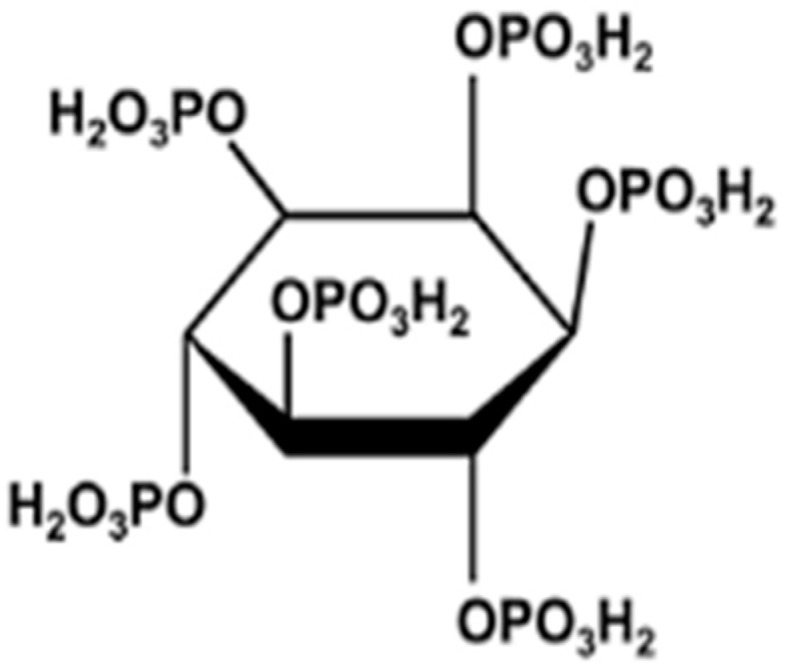
Molecular structure of phytic acid (Source: Higuchi et al., 2014) [20].

**Figure 2 ijms-20-04114-f002:**
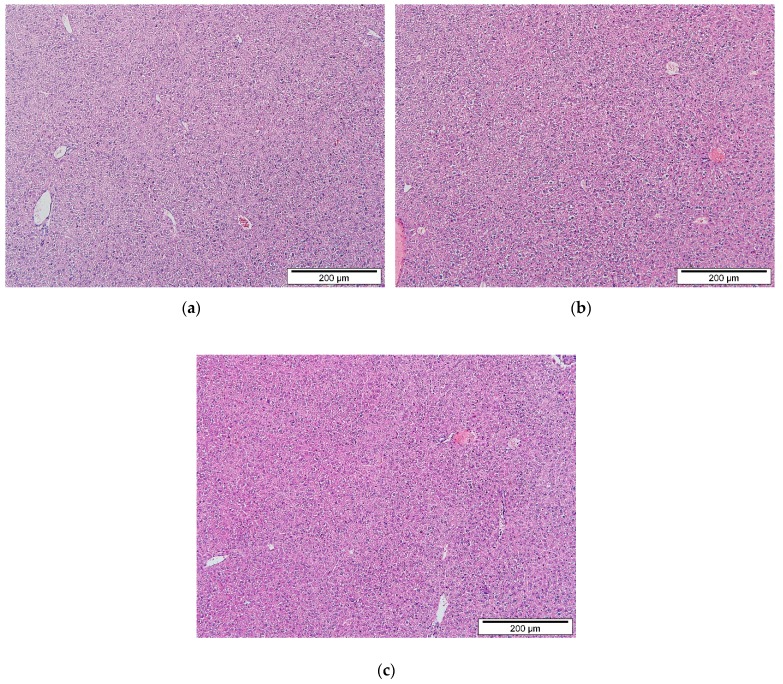
Liver section of: (**a**) control group of mice that received only deionized water; (**b**) 1000 mg/kg of IP_6_-CS-MNPs; (**c**) 2000 mg/kg of IP_6_-CS-MNPs (hematoxylin and eosin (H&E) stain, ×100).

**Figure 3 ijms-20-04114-f003:**
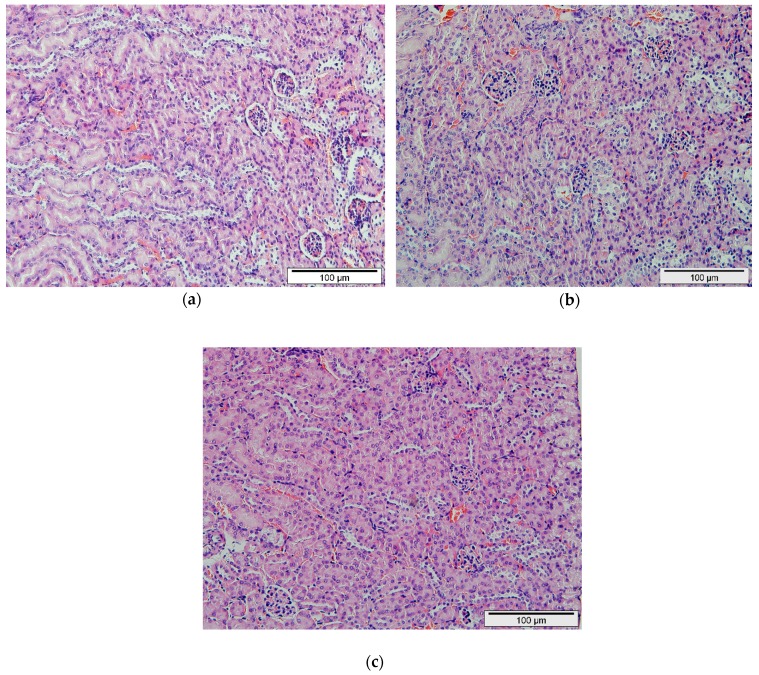
Kidney section of: (**a**) control group of mice that received only deionized water; (**b**) 1000 mg/kg of IP_6_-CS-MNPs; (**c**) 2000 mg/kg of IP_6_-CS-MNPs (H&E stain, ×100).

**Table 1 ijms-20-04114-t001:** Effect of phytic acid-chitosan-magnetic iron oxide nanoparticles (IP_6_-CS-MNPs) on mouse body weight.

Day		Body Weight (g)	
	Control	1000 mg/kg	2000 mg/kg
0	18.94 ± 0.55 ^a^	18.07± 0.44 ^a^	18.69 ± 0.45 ^a^
7	18.61 ± 0.54 ^a^	17.88 ± 0.40 ^a^	18.59 ± 0.44 ^a^
14	19.2 ± 0.52 ^a^	18.48 ± 0.30 ^a^	19.15 ± 0.47 ^a^

All values represent the mean ± standard error of mean (SEM). ^a^ Values in the same row with the similar superscript letter were not significantly different.

**Table 2 ijms-20-04114-t002:** Effects of IP_6_-CS-MNPs on relative organ weight.

Organ		Relative Organ Weight	
	Control	1000 mg/kg	2000 mg/kg
Liver	4.32 ± 0.25 ^a^	4.51 ± 0.38 ^a^	4.60 ± 0.19 ^a^
Kidney	0.64 ± 0.02 ^a^	0.65 ± 0.04 ^a^	0.57 ± 0.03 ^a^
Heart	0.71 ± 0.11 ^a^	0.52 ± 0.03 ^a^	0.53 ± 0.04 ^a^
Lung	0.69 ± 0.05 ^a^	0.72 ± 0.10 ^a^	0.92 ± 0.11 ^a^
Spleen	0.42 ± 0.10 ^a^	0.31 ± 0.03 ^a^	0.38 ± 0.05 ^a^
Brain	2.18 ± 0.08 ^a^	2.26 ± 0.05 ^a^	2.05 ± 0.52 ^a^
Colon	1.80 ± 0.26 ^a^	2.94 ± 0.89 ^a^	2.13 ± 0.15 ^a^

All values represent the mean ± SEM. ^a^ Values in the same row with similar superscript were not significantly different (*p* > 0.05).

**Table 3 ijms-20-04114-t003:** Effects of IP_6_-CS-MNPs on liver and kidney biochemical parameters.

Parameter	Control	1000 mg/kg b.w	2000 mg/kg b.w
ALP (U/L)	117.3 ± 5.68 ^a^	159.33 ± 6.73 ^b^	132.33 ± 6.51 ^a^
AST (U/L)	113.83 ± 20.46 ^a^	111.50 ± 13.60 ^a^	157.00 ± 26.10 ^a^
ALT (U/L)	23.67 ± 5.10 ^a^	18.67 ± 2.36 ^a^	20.17 ± 2.29 ^a^
Creat (μmol/L)	30.00 ± 1.21 ^a^	30.00 ± 0.82 ^a^	31.67 ± 1.33 ^a^
Urea (mmol/L)	8.02 ± 0.43 ^a^	7.67 ± 0.26 ^a^	9.75 ± 0.48 ^b^

All values represent the mean ± SEM. ^a^ Values in the same row with different superscript indicate they were significantly different (*p* < 0.05). ALP—alkaline phosphatase; AST—aspartate aminotransferase; ALT—alanine aminotransferase; Creat—creatinine.

**Table 4 ijms-20-04114-t004:** Animal grouping.

Groups	Dosage	Number of Mice
1	1 mL deionized water	6
2	1000 mg/kg b.w of IP_6_-CS-MNPs	6
3	2000 mg/kg b.w of IP_6_-CS-MNPs	6

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
