# Peer review of "The Acute Effects of Oral Administration of Phytic Acid-Chitosan-Magnetic Iron Oxide Nanoparticles in Mice"

_ijms, 2019, doi:10.3390/ijms20174114_

Round 1

Reviewer 1 Report

The authors have tested the therapeutic potential and possible side effects of an administration of a phytic acid-chitosan-magnetic iron oxide nanocomposite (IP6-CS-MNPs) under in vivo conditions in mice. They investigated animals up to 14 days after a single oral application of either 1000 or 2000 mg/ k.g. bw. of the nanocomposite.

While I like this scientific question about the effects of an administration of a supposed anti-cancer drug on body functions in mammals in vivo, the presented results are way to preliminary to make any sharp conclusions about the effects of IP6-CS-MNPs.

More specifically, I have the following questions to the authors:

1.) Is the nanoparticle indeed taken up by the cell tissue that was investigated here? This could be tested by labeling the nanoparticles with a tag and and visualizing them later post mortem in fixated tissue.

2. ) Is there any agglomeration of IP6-CS-MNPs at the concentrations used here? This could also be verified with labeled nanoparticles (see my point no.1)

3.) One important property of many nanoparticles is its potential to cross the Blood-brain barrier. If yes, an active compound (at relatively low concentrations) embedded in nanoparticles could reach neurons (and also tumors) inside the brain. So, did the nanoparticles enter cells within the brain? (see my point 1.)

4.) This might be a minor point, but since it is about the chosen title of the manuscript, I need to raise it here: The English writing needs extensive editing! For example, the title suggests to me that the authors did an "oral evaluation" (like an oral test, an oral interview, or an oral survey).
However, I guess, the authors mean something like "The acute effects of an oral administration of Phytic....". There are many more severe English language errors in the text. This should be corrected.  

Reviewer 2 Report

The authors report their results on the acute oral evaluation of phytic acid-chitosan-MNP prepared and characterized by some of them in a previous work. The manuscript need English language editing.

Some issues need to be addressed:

1) please don't refer to encapsulation for the surface functionalization with phytic acid.

2) a molecular structure formula of the acid could be more useful to the reader then a smple molecular formula in the text.

3) If the authors decided to reproduce the synthesis of the IP6-CS-MNPs with slight modifications of the original one, they should at least report some characterization (TEM images, size distribution, etc.). In nanoparticles synthesis even the slightest change can result in very different particles.

4) All the text from line 141 to line 164 cannot be considered discussion. It's a stating general consideration on already reported articles and should be moved to the introduction, if necessary, or cut.
